# Gene and Cell Therapy for Sarcomas: A Review

**DOI:** 10.3390/cancers17071125

**Published:** 2025-03-27

**Authors:** Sant P. Chawla, Skyler S. Pang, Darshit Jain, Samantha Jeffrey, Neal S. Chawla, Paul Y. Song, Frederick L. Hall, Erlinda M. Gordon

**Affiliations:** 1Sarcoma Oncology Center/Cancer Center of Southern California, Santa Monica, CA 90403, USA; santchawla@sarcomaoncology.com (S.P.C.); kaito.senuma@stud.semmelweis.hu (S.S.P.); darshitjain264@gmail.com (D.J.); sjeffrey@sarcomaoncology.com (S.J.); nealchawla@sarcomaoncology.com (N.S.C.); 2Aveni Foundation, Santa Monica, CA 90403, USA; 3NKGen Biotech, Inc., Santa Ana, CA 92705, USA; psong@nkgenbiotech.com; 4Delta Next-Gene, LLC, Santa Monica, CA 90405, USA; fhall@deltanextgen.com

**Keywords:** gene therapy, cell therapy, natural killer cell therapy, CAR-T therapy, virotherapy, sarcoma, solid tumors

## Abstract

In recent years, the number of FDA-approved gene and cell therapies for cancer has increased, albeit mostly for hematologic malignancies. This review focuses on gene and cell therapies for sarcomas that have been approved and therapies still in clinical trials. Rexin-G, the first and so far only tumor-targeted gene therapy product, was approved in the Philippines for all solid malignancies, including sarcomas, in 2007. Gendicine, the first oncolytic virus therapy, was approved in China for head and neck cancers, including sarcomas, in 2003. Afami-cel, an innovative T-cell therapy, was approved for synovial sarcoma in the United States in 2024. Today, there are many gene and cell therapy clinical trials for sarcomas, and the most promising studies are being discussed. The future holds great potential for gene and cell therapy for sarcomas, although high costs of vector production may be prohibitive. Overcoming this limitation is crucial for gene and cell therapy to become a widely accepted treatment for sarcomas—but saving lives, i.e., extending patient quality of life as well as overall survival, is worth it.

## 1. Introduction

Sarcomas are a rare group of neoplasms originating from mesenchymal cells, with over 50 known histological subtypes, including those arising from the bone, nerve, muscle, fat, blood vessels, and other connective tissues. The American Cancer Society predicted 17,370 new cases of sarcoma for 2023, along with 7280 related deaths [1]. The prognosis of metastatic/relapsing sarcomas remains poor, with a median survival rate of 10–12 months [2]. The heterogeneity of sarcomas, and resulting distinct sub-type specific characteristics, their high recurrence rates, and tendency for distant metastasis, continue to present significant challenges to providing optimal treatment [3].

The current clinical practice guidelines for localized sarcomas recommend surgical resection with clear oncological margins, and perioperative radiotherapy if unresectable. In the case of the relapsing/metastatic setting, doxorubicin-based chemotherapy regimens are preferred as the first-line treatment [4]. However, concerns over systemic toxicity and the limited long-term success rates have driven research towards more targeted approaches. Subsequent lines of therapy have evolved to include innovative approaches, including receptor tyrosine kinase inhibitors, new chemotherapy drugs (such as eribulin, a microtubule targeting agent, and trabectedin, an alkylating agent), immunotherapy with gene expression vectors and immune checkpoint inhibitors (ICI), virotherapy, gene therapy, and cell therapy. This review explores the emerging role of gene and cell therapies in the treatment of sarcomas and discusses potential future directions in this rapidly advancing field.

### Search Strategy

A comprehensive literature search was conducted utilizing the following databases: PubMed, Medline, Google Scholar and clinicaltrials.gov. Search terms included “gene therapy”, “cell therapy”, “NK cell therapy, “CAR-T therapy”, “virotherapy”, “sarcoma”, “gene therapy”, and “solid tumors”. Additional sources were identified through manual searching for references of relevant studies. No language restrictions were set. The NCT number, study status, condition, and phase were noted for clinical trials.

## 2. Retrovirus-Based Gene Therapy

The concept of strategic gene transfer as a form of therapy first took shape during the 1960s and 1970s, spurred by research demonstrating the feasibility of inserting foreign deoxyribonucleic acid (DNA) into cells to confer new, permanent, and functioning genes [5]. This period saw the discovery of papovaviruses, simian virus 40, and polyomaviruses, which embed genetic material into host cells to induce virus reproduction, which may result in oncogenic transformation. These findings provided clinical insights into the molecular mechanisms of viral subversion, proto-oncogene activation, and genetic transformation [6,7]. In 1972, Theodore Friedman proposed the idea that genetically engineered tumor viruses could serve as vectors for delivering therapeutic genes to treat inherited diseases [8].

In 1984, William French Anderson, known as the “father of gene therapy”, addressed the prospects of utilizing retroviral vectors in gene therapy by successfully implementing functional genes into mouse bone marrow cells [9,10]. In 1990, Anderson carried out the first-ever clinical trial for gene therapy, utilizing a murine retroviral vector to introduce the adenosine deaminase (ADA) gene into T cells of children with severe combined immunodeficiency (SCID) [11]. Although one of the participants demonstrated restoration in immune function and lives to this day, the trial, along with subsequent studies, raised many concerns, particularly in terms of safety. Key issues included the risk of producing replication-competent viruses and the potential for insertional mutagenesis, which could promote tumorigenesis during cell engraftment [12]. These concerns underscored the urgent need for safe and precise gene delivery systems.

From the mid-1990s to the early 2000s, the focus of oncology research shifted substantially to targeted gene delivery, which has long been considered the “Holy Grail” of cancer gene therapy. A significant development in this area was the incorporation of target-specific ligands into viral vectors allowing the virus to display selected ligands on its capsid or envelope surface, facilitating targeted cell recognition [13] (Figure 1). This versatile technique was pioneered by Stephen J. Russell, who developed single-chain antibodies expressed on the surface of enveloped viral particles and can be applied to various viral vectors, playing a crucial role in advancing targeted gene therapy [14].

### 2.1. Targeting Executive Oncogenic Drivers

In the early 1990s, multiple researchers published findings linking cyclins to oncogenesis [16]. Human cyclin G1 (*CCNG1*), in particular, garnered interest following the work of Frederick L. Hall et al. in 1994 who first characterized the human *CCNG1* gene via molecular cloning and demonstrated overexpression of the cyclin G1 oncogene that was independent of the p53 tumor suppressor in osteosarcoma cell lines [16].

In 2000, Gordon et al. developed and investigated the physiological performance of a retroviral expression vector as a tumor-targeted gene delivery system, expressing a cytocidal dominant-negative construct of cyclin G1 (dnG1) in a metastatic liver model in nude mice [17], and in 2001, Gordon et al. showed that intravenous delivery of the mutated *CCNG1* gene vector induced shrinkage of tumors in a nude mouse model of metastatic cancer, confirming cell death via apoptosis-mediated pathways [18].

### 2.2. Rexin-G, the First Clinical Tumor-Targeted Gene Therapy Vector for Sarcoma

The clinical development of Rexin-G (Mx-dnG1, dnG1) represents the first and, so far, only tumor-targeted gene therapy to receive accelerated approval by a federal agency in 2007. Rexin-G utilizes a murine leukemia virus-based retroviral vector platform engineered with a von Willebrand factor-derived collagen binding decapeptide on its outer envelope which enhances partitioning and tumor microenvironment specificity [19].

A phase 1/2 clinical trial using Rexin-G monotherapy for heavily pre-treated chemo-resistant sarcomas was performed with the objectives of determining the safety and antitumoral activity by examining adverse events and identifying a definitive anti-tumor response. Thirty-six advanced sarcoma patients received five escalating dose levels.

In 33 evaluable patients who were assessed for tumor response using Response Evaluation Criteria in Solid Tumors (RECIST) v1.0, International Positron Emission Tomography (PET), and CHOI criteria, the most favorable survival outcomes were observed in those receiving the highest doses of Rexin-G. All serious adverse events experienced were reported to be related to the disease and deemed unrelated to the drug. The majority of the drug-related events reported were of grade 1–2 severity including fatigue, hypersensitivity, and chills [20]. This phase 1/2 study demonstrated antitumor activity with a wide margin of safety for Rexin-G. Moreover, this study, as well as a phase 1/2 study using Rexin-G for advanced pancreatic cancer, established a dose-dependent increase in overall survival advantage [20,21].

### 2.3. DNG64, a Paradigm Shift in Targeted Gene Delivery

DNG64 (also known as DeltaRex-G) is an off-the-shelf replication incompetent Chimeric Amphotropic RNA Vector (CAR-V) equipped with a navigational system, a SIG-binding peptide, and encodes a Cyclin G1 inhibitor (Figure 2). Injected intravenously, DNG64 nanoparticles seek out and bind to abnormal Signature (SIG) proteins exposed in the TME, which augments effective vector concentration in tumors. DNG64 kills not only cancer cells and tumor-associated vasculature but also kills tumor-associated fibroblasts and reduces stroma production which facilitates entry of immune cells, cancer drugs and immune checkpoint inhibitors into the TME. This targeted approach to gene delivery circumvents the off-target adverse events commonly seen in conventional chemotherapy and immune checkpoint inhibitors (ICIs) and introduces a new paradigm in the treatment of sarcomas and other cancer types.

Expanded access for DNG64 is currently available for an intermediate-size population of advanced sarcomas, pancreatic adenocarcinoma and carcinoma of the breast (NCT04091295). The United States Food and Drug Administration Center for Biologics Evaluation and Research (USFDA-CBER) authorized its use in these cancer types based on repeated demonstrations of safety and efficacy in phase 1/2 studies with long-term (>10 years) survivors after DNG64 treatment initiation [22]. These studies also paved the way for the potential of using DNG64 in the perioperative setting to prevent recurrence and metastasis [23].

Mechanistically, inhibiting *CCNG1* disrupts the Cyclin G1/Cdk2/Myc/Mdm2/p53 Axis of executive oncogenes, a critical regulatory pathway governing cell survival, growth, and proliferation, inducing cell cycle arrest and apoptosis of cancer cells [22,23]. Figure 3 illustrates the mitogenic signal transduction via proline-directed protein phosphorylation pathways involved with the expression and overexpression of the *CCNG1* oncogene. In collaboration with BostonGene Corp., Waltham, MA, USA retrospective studies of *CCNG1* expression levels in archived tumor samples were conducted, which confirmed enhanced *CCNG1* expression in all cancer types tested including sarcomas, urethelial carcinoma, testicular carcinoma, pancreatic adenocarcinoma, and carcinoma of breast [24].

Recently, *CCNG1* expression was further evaluated in 179 sarcoma tumors, and the results of DNG64-treated patients with known *CCNG1* expression levels were reported [25]. In this study, 21% showed high *CCNG1* expression, 35% showed medium-high *CCNG1* expression, 40% showed medium-low *CCNG1* expression, and 4% showed low expression. Nine patients with metastatic sarcomas (subtypes included osteosarcoma, small round cell desmoplastic tumor, chondrosarcoma, malignant peripheral nerve sheath tumor, uterine leiomyosarcoma, and dedifferentiated liposarcoma) were treated with either DNG64 monotherapy or DNG64+ (DNG64 with an FDA approved drug) and evaluated for overall survival, duration of treatment, and disease responses via RECIST v1.1 criteria.

The majority of patients had previously progressed with standard doses of chemotherapy. No serious treatment-related adverse events were reported. The median survival of sarcoma patients treated with DNG64+ has not been reached, and to date, is at least >13 months, equivalent to the median survival of patients treated with first line standard (doxorubicin) therapy [2,4]. The results indicated that DNG64 may evoke tumor growth stabilization in patients who previously failed standard chemotherapy and may prime tumors to better respond to other treatments including chemotherapy, targeted therapy, and immunotherapy.

Phase 2 randomized studies with an expanded sample size are planned for 2025 to further evaluate the efficacy and safety of DNG64+ and to correlate *CCNG1* expression level and circulating tumor DNA (ctDNA) level with treatment outcome parameters in response to DNG64+ gene therapy.

### 2.4. The Genevieve Protocol: Rexin-G and Regulatable Reximmune-C

The Genevieve Protocol is a personalized vaccination strategy aimed specifically at enabling immune cell trafficking in the tumor microenvironment (TME) and evoking immune responses against a patient’s own specific cancer to improve tumor control and survival [26].

This strategy is a dual-targeted gene therapy regimen, combining (1) Rexin-G, a tumor-targeted retrovector encoding a cytocidal gene to destroy the cancer cells and tumor vasculature and expose tumor neoantigens, and (2) Reximmune-C, a tumor-targeted retrovector encoding a regulatable GM-CSF/Herpes simplex virus Type 1 thymidine kinase (HSV-tk) gene for local paracrine secretion. Reximmune-C polarizes M1 to M2 macrophages, matures dendritic cells, activates T cells, and recruits the patient’s own cytotoxic T cells into the TME for in situ tumor neoantigen recognition.

In this phase 1/2 study, 16 patients with various advanced cancers including osteosarcoma, liposarcoma and Ewing sarcoma were treated. The median progression-free survival (PFS) was 13 months and the median overall survival was >21 months with one patient with locally advanced osteosarcoma still alive to date.

### 2.5. GEN2

Like Reximmune-C, “GEN2 is a non-replicating off-the-shelf gene therapy vector product being developed as a cancer immunotherapy to activate a patient’s immune system against their personal cancer antigens (neoantigens). The vector payload encodes for a suicide gene, an enhanced viral thymidine kinase enzyme (HSV-eTK), which in the presence of a prodrug, valganciclovir, causes the tumor to release patient-specific tumor antigens. These neoantigens, in the presence of a human immune modulator cytokine, granulocyte-macrophage colony-stimulating factor (hGM-CSF), result in the generation of immune effector cells”. The safety and efficacy of GEN2 are currently being tested in a phase 1 study for all solid tumors (NCT06391918) [27].

### 2.6. Points to Consider with Retroviral-Based Gene Therapy

Gamma-integrating Type C viruses have the risk of insertional mutagenesis, development of vector-neutralizing antibodies, recombination events, and delayed long-term adverse events. Any study using gamma-integrating Type C viruses would require testing for short, delayed and long-term adverse events.

To date, only one retroviral-based gene therapy product, Rexin-G, has received marketing approval in the Philippines as monotherapy for all solid malignancies including sarcoma. The risks of retroviral gene therapy have been mitigated by the development of a targeted gene delivery system wherein off-target cells are spared. No vector-neutralizing antibodies nor non-target DNA integration was detected in phase 1/2 studies using Rexin-G [20,21]. Long-term expression of the therapeutic transgene that might cause second malignancies is not expected since these cytocidal vectors are not designed for continued expression, and target cancer cells, tumor-associated vasculature and stroma-producing fibroblasts are eradicated. In fact, long-term adverse events and the development of cancer have not been reported in long-term cancer survivors treated with tumor-targeted gene vectors [20,21,22,26].

On the other hand, research and development of retroviral-based vectors, vector production, lot release testing and clinical trials are costly. Economies of scale would reduce the cost of retroviral vector production, resulting in a product cost that is expected to be equivalent to that of immunotherapy products.

## 3. Oncolytic Virotherapy/Gene Therapy

Oncolytic virotherapy (OVT) refers to the use of replication-competent viruses as therapeutic agents that selectively target and destroy cancer cells. In the process, these viruses trigger a strong systemic immune response by releasing viral progeny, tumor-associated antigens, danger signals, and proinflammatory cytokines, which can enhance the body’s overall antitumor immunity. However, the high immunogenicity limits repeated dosing and shortens therapeutic gene expression. Pre-existing immunity from natural adenovirus exposure also hinders the efficacy of OTV.

To address these challenges, strategies include using rare adenoviral serotypes (e.g., Ad26, Ad35), non-human adenoviruses (e.g., chimpanzee-derived), and helper-dependent vectors with reduced viral genes. Although the immunogenicity of OTV poses challenges, it also enhances immune activation against tumors, making oncolytic viruses effective for immunotherapy and combination strategies in sarcoma treatment [28].

### 3.1. Herpes Simplex Virus Type 1 Oncolytic Virus

The United States Food and Drug Administration (USFDA)-approved oncolytic virus therapy is a genetically modified herpes simplex virus (HSV-1), Talimogene laherparepvec (T-VEC) expressing a GM-CSF transgene approved for melanoma. In addition, the results of a recent study using Talimogene laherparepvec, in combination with nivolumab and trabectedin (TNT regimen) for previously treated advanced sarcomas, showed promising results [29]. In this phase 2 study, intratumoral talimogene laherparepvec (1 × 10^8^ plaque-forming units/mL) was given q 2 weeks, nivolumab (3 mg/kg) i.v. q 2 weeks, and trabectedin (1.2 mg/m^2^) continuous intravenous (CIV) infusion q 3 weeks. In 39 evaluable patients, the median progression-free survival was 7.8 months compared to 4.1 months with trabectedin alone, and the median overall survival was 19.3 months, which is similar to doxorubicin standard first-line therapy [2,4]. One patient with alveolar rhabdomyosarcoma was able to undergo a complete surgical resection after two months of therapy and has not experienced a recurrence after 3 years with no further cancer therapy. Treatment-related adverse events included anemia, thrombocytopenia, neutropenia, increased alanine transaminase, decreased left ventricular ejection fraction, dehydration, and hyponatremia. The authors concluded that the TNT regimen is effective with manageable toxicity for previously treated advanced sarcomas.

### 3.2. Adenoviral-Based Oncolytic Virus

Adenoviruses (Ad) are among the most favored oncolytic viruses due to their high stability, ease of production, non-integration into the host genome, large packaging capacity around 35 kilobase (kb), and favorable safety profile in clinical trials.

#### 3.2.1. Gendicine

In the year 2000, an adenovirus-mediated p53 gene therapy, named Ad5p53, was introduced. Given that mutations in the tumor suppressor protein p53 are common in sarcomas, this therapy employed a replication-deficient adenovirus, with the early region 1A (E1A) gene deleted, to introduce wild-type p53 cDNA [30]. The gene is controlled by a cytomegalovirus promoter and includes a Flag tag to facilitate the reactivation of functional p53 [31]. Preclinical in vitro and in vivo studies demonstrated the ability of Ad5p53 to slow and delay tumor growth.

In 2003, the Chinese FDA approved Gendicine, a recombinant human p53 adenovirus designed for local intratumoral injection [31]. A study assessed its efficacy in 12 patients with uterine sarcoma, where recombinant adenovirus-p53 (rAd-p53) was administered alongside chemotherapy including bleomycin, cisplatin, epirubicin, and isocyclophosphamide. The findings revealed that one patient experienced complete remission (CR), seven showed partial responses (PR), three had stabilized disease (SD), and one patient’s condition worsened with disease progression. The overall response rate (CR + PR) was 66.7%, with a disease control rate (CR + PR + SD) of 91.7%. The median overall survival (OS) was 24 months, with two long-term survivors: one remained tumor-free for 60 months, while the other had progression-free survival of 39 months, and survived for 53 months with slowed disease [31]. Through over 12 years of commercial use, when used alongside chemotherapy and radiotherapy, Gendicine has achieved significantly higher response rates compared to standard treatments. Findings from 13 published studies, including long-term survival data, demonstrate that combining Gendicine leads to longer progression-free survival compared to standard therapies on their own [32].

Another study employed Gendicine in treating 71 patients with advanced soft tissue sarcomas, including Ewing sarcoma [33]. The study implemented two treatment arms; 36 patients received Gendicine in combination with radiotherapy and hyperthermia, and 35 patients received radiotherapy and hyperthermia (control arm). The Gendicine arm had a disease control rate (DCR) of 83.33% versus 54.29% for the control arm. Additionally, improvements in both overall survival and progression-free survival rates were observed in the Gendicine arm. These results underscore the potential of adenovirus-p53 gene therapy in restoring tumor-suppressing functions, offering another possible treatment option for sarcomas.

#### 3.2.2. AdAPT-001

An ongoing phase 1/2 trial (Beta Prime) for sarcomas (soft tissue sarcoma and chondrosarcoma) utilizes AdAPT-001, an oncolytic replicative type 5 adenovirus armed with a Transforming Growth Factor (TGF)-β receptor-immunoglobulin Fc fusion trap, designed to neutralize isoforms 1 and 3 of the profibrotic and immunosuppressive cytokine, TGF-β (NCT04673942) [34]. The immunogenic oncolytic virus is injected intratumorally to evoke an inflammatory reaction resulting in overexpression of TGF-β in the tumor microenvironment. In this study, a total of 36 patients were enrolled; 12 received AdAPT-001 monotherapy and 24 received combination therapy with an ICI. The primary endpoints were objective response rate (ORR) and PFS. Results showed that the combination of AdAPT-001 with an ICI achieved a 29.1% overall response rate (ORR, 7/24 patients), including one CR and six PR. The clinical benefit rate (CBR) for the combination was 62.5%. Monotherapy with AdAPT-001 resulted in a 22.2% ORR and a 44.4% CBR. Adverse events were generally mild, with only one immune-related adverse event (hypophysitis) reported. The combination therapy demonstrated promising efficacy and was well tolerated, particularly in ICI-refractory patients. PFS was 3.5 months [35]. This strategy appears to reverse resistance to ICIs and induce both local and systemic anti-tumor responses. Further research is ongoing to identify biomarkers that could predict which patients are most likely to benefit from this combination therapy [36].

### 3.3. Points to Consider with Oncolytic Viruses

To date, there is only one oncolytic virus, Gendecine, that has been approved in China for sarcoma. This treatment was approved when given intratumorally in combination with intravenous chemotherapy. The efficacy of oncolytic viruses is impeded by the physical barriers in the tumor microenvironment that prevent OVs from reaching the target cancer cells, the development of antiviral antibodies and the possibility of systemic infection. Current attempts at addressing these hurdles include manipulations of the virus to create more potent OVs without increasing toxicities as well as combining OVs with FDA-approved immune checkpoint inhibitors. Like retroviral-based vectors, oncolytic vector production is expensive but large-scale manufacturing would, in time, reduce the cost to that of immunotherapy products currently in use.

## 4. Cell Therapy for Sarcomas

Cell therapy is a therapeutic approach that involves the use of cells to treat various forms of cancer. According to the American Society of Gene and Cell Therapy (ASGCT), cell therapies can be categorized into several subtypes such as stem cell therapy, TIL therapy, and the “other” subtype. Natural killer (NK) cell therapy, chimeric antigen receptor T-cell (CAR-T) and T Cell Receptor (TCR) therapy fall under the “other” category, with emphasis on modifying the anti-tumoral responses of immune cells [37].

### 4.1. Natural Killer Cell Therapy

NK cells are lymphoid-derived immune cells of the innate immune system, known for their ability to target and destroy virally infected cells, diseased cells and tumors. These cells possess a unique array of activating and inhibitory receptors on their cell surface, enabling them to recognize viral proteins, stress-induced ligands, and decreased Major Histocompatibility Complex (MHC) class I-expressing molecules. Upon cell recognition, NK cells induce apoptosis in target cells through the granzyme/perforin pathway or via the death receptor–ligand pathway (tumor necrosis factor family) [38,39].

#### Autologous Enhanced Natural Killer Cell Therapy (SNK01)

SNK01 cell therapy is a first-in-kind, autologous, non-genetically modified NK cell product with significant anti-tumor cytotoxicity and over 90% expression of cell determinants CD16, NKG2D, NKp46, and DNAM-1, that can be consistently produced even from chemotherapy-treated cancer patients [40]. While NK cell therapy has shown promising results in treating hematologic malignancies, its efficacy against solid tumors has been limited. This limitation is attributed to its inability to traffic through the solid tumor microenvironment efficiently, as immunosuppressive cytokines inhibit the NK cells at the site of the tumor [41] (Figure 4).

To enhance the effectiveness of NK cell therapy against solid tumors, researchers have explored the combination of NK cells with ICIs such as programmed death 1 (PD-1) or programmed death ligand 1 (PD-L1) inhibitors [38]. The addition of an ICI was believed to lead to a synergistic effect for immune cell trafficking and increased anti-tumor activity [41]. Interestingly, integrating natural killer cells with ICIs may improve the responsiveness of PD-L1-negative tumors to these therapies [42].

Studies have investigated the combined synergistic effect of ICIs and NK cell therapy in sarcomas. Two case reports of patients with chemo-resistant metastatic sarcomas detailed the outcomes for combined SNK01 and pembrolizumab. Pembrolizumab is an anti-PD-1 monoclonal antibody. As PD-1 is expressed in T cells, the binding of pembrolizumab to T cells exposes the tumor to a T-cell-mediated immune response [43].

In one case, a 32-year-old patient with metastatic desmoplastic small round cell tumor received five weekly doses of SNK01. Despite his extensive liver metastases and abdominal and pelvic disease, the patient had stable disease in his primary tumor. The combination of SNK01 and pembrolizumab was authorized by the USFDA and was then given as a single patient investigational new drug application. The patient achieved a 47% partial response by RECIST v1.1 after one year of treatment, followed by sustained remission with no evidence of disease after subsequent surgery, radiation, and chemotherapy. His last scan in the following year showed sustained remission with no evidence of disease [43].

In another case, a 57-year-old male with high-grade pleomorphic liposarcoma of the right pelvis and PD-L1-positive chondrosarcoma received SNK01 and pembrolizumab. The patient achieved a 38% partial response after four months of treatment but unfortunately died from post-operative sepsis following a debulking surgery. He received 18 cycles of combined therapy and survived 12 additional months [43].

Recently, a phase I study (NCT03941262) was initiated to investigate the combination of SNK01 and avelumab in advanced refractory sarcomas. Avelumab blocks PD-L1 from interacting with PD-1 and B7.1, a co-stimulatory molecule on antigen-presenting cells and T cells, thus activating both innate and adaptive immune responses. The interim analysis of the study showed 15 enrolled patients with advanced sarcomas. The objective response by RECIST 1.1 included a partial response in two patients and (ORR of 13.3%) and stable disease in three patients [44]. Overall, the treatment was well tolerated and showed clinical activity in advanced sarcomas independent of PD-L1 status.

These findings underscore the potential of NK cell therapy in combination with immunotherapy for treating solid tumors. Further combinations are being explored for the synergistic effects of DNG64 in NK cell therapy. Gordon et al. have discussed the potential of combining SNK01 with DNG64, a tumor-targeted retroviral vector that binds to anaplastic SIG proteins exposed by invading tumors in the tumor microenvironment [43]. DNG64 not only kills cancer cells, but also tumor-associated neovasculature and tumor-associated fibroblasts, thereby reducing extracellular matrix production. This would improve NK cell trafficking in the TME, further amplifying the anti-tumoral activity of NK cells.

### 4.2. Mesenchymal Stromal Cell Therapy

Mesenchymal stromal cell (MSC) therapy is an emerging approach in bone sarcomas, particularly for treating osteosarcoma and Ewing sarcoma [45]. MSCs are capable of multipotent differentiation and can deliver anti-tumoral agents directly to cancerous tissues. MSCs are attracted to the tumor-associated inflammatory signals within the tumor microenvironment and invite themselves to the tumor [46]. In particular, genetically engineered MSCs expressing certain proteins could offer new treatment avenues. For example, MSCs engineered to express tumor necrosis factor-related apoptosis-inducing ligand (TRAIL) can activate the extrinsic apoptosis pathway in cancer cells [47]. Osteoprotegenin, a decoy receptor for receptor activator of nuclear factor kappaB ligand (RANKL), could inhibit bone resorption and osteoclast activity, blocking osteosarcoma progression associated with increased RANKL release [48]. Other approaches include MSCs delivering cytosine deaminase-5-fluorouracil (CD-5FC), a prodrug capable of converting cytosine into uracil together with a cytotoxic drug, inducing cancer apoptosis [49]. Interleukin (IL)-12 delivered locally can increase fas expression in tumors, also leading to tumor apoptosis [50].

### 4.3. Chimeric Antigen Receptor (CAR)-T Cell Therapy

CAR-T cell therapy is a fascinating, newly approved form of adoptive T cell therapy. Unlike other designs that involve genetic engineering of the patient’s existing T-cell receptors, CAR-T therapy utilizes a fully synthetic chimeric antigen receptor (CAR) on the cell surface to enhance T-cell specificity [51] (Figure 5). The chimeric nature of CAR-T cells arises from the fusion of a single-chain variable fragment (scFv) of the binding domain of a monoclonal antibody with an intracellular (IC) T-cell activation domain. This design allows surface antigen recognition without depending on MHC molecule antigen presentation. Examples of IC T-cell activation domains include CD28, 4-1BB, OX40, and CD3ζ [52]. To date, five generations of CAR-T therapies have been developed, each distinguished by the incorporation of various costimulatory receptor signaling domains [53].

**Figure 5 cancers-17-01125-f005:**
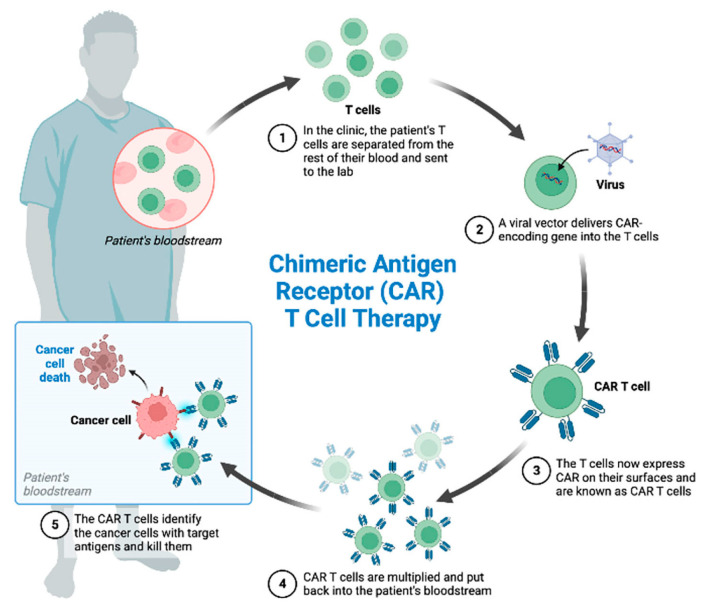
CAR-T mechanism of action. CAR-T cell therapy consists of collection of the patient’s T cells by leukapheresis, subsequent engineered manipulation by adding a gene for a receptor (called a chimeric antigen receptor, “CAR”), ex vivo expansion and reinfusion. Each CAR is produced to target a specific cancer cell antigen [54].

CAR-T cell therapy has shown significant success in hematologic malignancies with multiple FDA-approved CAR-T cell therapies including Tisaglenlecleucel and brexucabtagene autoleucel for acute lymphoblastic leukemia, axicabtagene ciloleucel for large b-cell lymphoma, and idecaptagene vicleucel for multiple myeloma [55]. What sets CAR-T apart from other therapies is its remarkable specificity. The synthetic CAR allows for the recognition of specific antigens on the surface of tumor cells, unlike ICIs or cancer vaccines, which rely on the patient’s immune system and often struggle with specificity and efficacy [56]. This is especially effective in hematologic malignancies, as certain CDs are only seen in leukemias and lymphomas [57].

#### 4.3.1. Afamitresgene Autoleucelis (Afami-Cel; TECELRA)

Afamitresgene autoleucelis is currently the only USFDA approved CAR-T therapy for metastatic or unresectable synovial sarcoma [58]. This therapy involves genetically modified T cells which express a TCR specific for the human melanoma-associated antigen 4 (MAGE-A4), a cancer-testis antigen overly expressed in synovial sarcoma. For this therapy to be effective, patients must be positive for Human Leukocyte Antigen (HLA) -A02:01P, -A02:02P, -A02:03P, or -A02:06P, as these HLA alleles are necessary for the translation of the targeted MAGE-A4 peptide sequence GVYDGREHTV. Binding to MAGE-A4 expressed cells induces T-cell mediated lysis and T-cell proliferation [59,60].

The approval of afamitresgene autoleucelis was based on a phase 2, single-arm, open-label study (ADP-0044-002 Cohort 1) named SPEARHEAD-1, which provided substantial evidence for both safety and efficacy. Cohort 1 recruited patients aged 16–75 with metastatic or unresectable synovial sarcoma or myxoid round liposarcoma. They were screened for the aforementioned HLA alleles and MAGE-4 expression [60]. T cells were obtained through leukapheresis and modified to express the MAGE-A4 targeted T cell receptors. Before receiving the Afami-cel infusion, patients underwent lymphodepletion chemotherapy, consisting of fludarabine and cyclophosphamide. A total of 52 patients received Afami-cel. The clinical trial had an overall response rate of 39% (17 of 44 patients) in patients with synovial sarcoma. Though all patients were found to have treatment-emergent adverse effects of lymphopenia, neutropenia, and leukopenia, no grade 3 adverse events were reported [59].

#### 4.3.2. HER2-Specific CAR-T Cell Therapy

NCT00902044 evaluates autologous human epidermal growth factor receptor 2 (HER2)-specific CAR-T therapy for patients with advanced sarcomas [61]. HER2, an epidermal growth factor receptor primarily targeted in breast cancer, shows moderate expression in sarcomas. HER2-targeted therapies have been used previously, including trastuzumab with chemotherapy against HER-2 positive metastatic osteosarcomas, and implementing HER-2-specific CAR-T for progressive glioblastoma [62,63]. CAR-T therapy in this study aims to target cell surface HER2 receptors effectively. Fourteen patients received HER2-specific CAR-T therapy following lymphodepletion, resulting in three patients achieving CR, four cases of SD, and seven cases of progressive disease (PD). Cytokine release syndrome (CRS) was reported in 11 patients, though no neurological adverse events occurred [61].

NCT04995003 is another ongoing recruiting trial where HER2-CAR-T cells are infused in combination with an ICI (pembrolizumab or nivolumab) [64].

#### 4.3.3. EGFR-Specific CAR-T Cell Therapy

NCT03618381 (active, recruiting) includes epidermal growth factor receptor (EGFR)-specific CAR-T cells (Arm A) and EGFR and CD19-specific CAR-T cells (Arm B) in patients with osteosarcoma, Ewing sarcoma, rhabdomyosarcoma, synovial sarcoma, and clear cell sarcoma [65].

Other CAR-T cell therapies for sarcomas include NCT03635632. This trial involves CAR-T cells targeting GD2, a tumor-associated antigen commonly found in neuroblastoma and osteosarcoma. The cells were further modified with C7R, a gene capable of inducing constitutive cytokine release. The trial is in progress and is currently active, not recruiting [66,67]. NCT03373097 is another ongoing trial involving CAR-T cells targeting GD2 whereas NCT03721068 is a similar trial that includes CAR T-Cells targeting GD2 with IL-15+ iCaspase9. Both these trials are active, recruiting [68,69].

While CAR-T cell therapy shows great promise, several challenges must be overcome. Patients may experience cytokine release syndrome, neurotoxicity, and on-target and off-target effects, which can potentially lead to life-threatening toxicities [56]. Proposed strategies to mitigate these events include tuning the affinity of the CAR scFv to recognize tumors while sparing non-tumorous cells, the use of clinically approved drugs to lessen toxicity, or iCaspase-9 associated suicide switches to eliminate the CAR-T cells in life-threatening situations [70]. A recent publication by Haroun and Gordon provided the rationale for the use of DNG64 in steroid-resistant CRS. The hypothesis is that a brief administration of DNG64 would kill a certain proportion of CAR-T cells, reduce IL-6, and hence, the severity of CRS while maintaining the efficacy of unaffected T cells [71]. Resistance remains a challenge, as antigen loss, immunosuppressive environments, tumor barriers, and T-cell exhaustion impair CAR-T persistence and efficacy [70].

The high cost and limited scale manufacturing of CAR-T cells also remains a large limitation [72]. Overcoming these limitations is critical for CAR-T cell therapy to become a widely accepted treatment for solid tumors and sarcomas.

### 4.4. T Cell Receptor (TCR) Therapy

TCR therapy is an alternative adoptive T-cell therapy that has been used in multiple clinical trials. Unlike chimeric antigen receptor (CAR)-T therapy, which recognizes tumor cell surface antigens, this treatment enables the detection of intracellular peptides.

#### Letetresgene Autoleucel (Lete-Cel, GSK3377794) Therapy

One notable study (NCT01343043) examined the safety and efficacy of Letetresgene autoleucel, a therapy involving the genetic modification of autologous T cell receptors [73]. The transduced TCR recognizes New York esophageal squamous cell carcinoma-1 (NY-ESO-1), in conjunction with HLA-A*02 alleles. Among 12 patients with advanced synovial sarcoma, a 50% overall response rate was observed, with one CR, five PR, five SD, and one PD. This targeted therapy is particularly promising in overcoming the immune resistance mechanisms in synovial sarcoma, which are capable of inhibiting immune infiltration.

NCT03967223 (active, not recruiting) evaluated the efficacy and safety of Letetresgene autoleucel (lete-cel) in patients with advanced/metastatic synovial sarcoma or myxoid/round cell sarcoma. Lete-cel therapy employs an engineered T cell targeting a tumor-associated protein with high expression in multiple tumor types, known as the New York esophageal antigen-1 (NY-ESO-1) [74,75].

D’Angelo et al. recruited patients for two substudies, both of which include NY-ESO-1 positive patients with advanced/metastatic synovial sarcoma or myxoid/round cell liposarcoma. Substudy 1 recruited previously untreated patients, whereas substudy 2 recruited patients previously treated with anthracycline-based chemotherapy, who had progressed on prior treatment, and who are HLA -A*02:01, *02:05, or *02:06-positive. Patients underwent apheresis for lete-cel manufacturing, followed by lymphopletion using fludarabine and cyclophosphamide, after which patients received Lete-cel infusion [74,76].

D’Angelo et al. reported the planned interim analysis of substudy 2 in May of 2024. Forty-five patients were evaluated for efficacy, with an ORR of 40% (2 CR, 16 PR). The median duration of response was 10.6 months, though 12 of 18 patients were censored. Seventy-three patients were evaluated for safety; common adverse events included cytokine release syndrome (89%), neutropenia (73%), thrombocytopenia (63%), rash (53%), anemia (52%) and leukopenia (49%). Serious adverse events included cytopenia (86%), cytokine release syndrome (12%), and rash (23%) [75].

### 4.5. Dendritic Cell Therapy

Several clinical trials also involve dendritic cells, antigen-presenting cells crucial for both the innate and adaptive immune system. NCT00405327 and NCT02496520 focus on autologous dendritic cell vaccines to treat metastatic and relapsing sarcomas [77,78]. NCT01291420 utilizes autologous dendritic cells that are RNA-modified to express WT-1 proteins for sarcomas and various solid tumors [79]. NCT00365872 is a phase II trial combining intratumoral injections of dendritic cells with external beam radiation therapy; a phase I study showed promising long-term outcomes in soft tissue sarcomas [80]. NCT00001566 administers a peptide-pulsed dendritic cell vaccination for pediatric patients with metastatic sarcomas like Ewing’s sarcoma and rhabdomyosarcoma [81]. Trials such as NCT01803152 and NCT00923351 also utilize autologous dendritic cell vaccination across various types of sarcomas [82,83].

### 4.6. Points to Consider with Gene Modified Cell Therapy

To date, only one CAR-T cell therapy, Afamitresgene autoleucelis, for advanced or metastatic synovial sarcoma, is approved in the United States. Therapies such as CAR-T and TCR therapy that involve a lymphodepleting procedure are frequently associated with significant adverse events such as cytopenias, cytokine release syndrome and neurotoxicities [56]. Further, patients have to wait 3–6 weeks to receive these gene-modified T and NK cells while their cancers are progressing. Strategies to treat the adverse reactions include the use of steroids, tocilizumab, and supportive care. Proposed strategies include enhancing the tumor specificity of CAR-T cells, and in the case of CRS, using iCaspase-9 associated suicide switches to eliminate the CAR-T cells in life-threatening situations [70] or using DNG64 in steroid-resistant CRS [71]. The high cost and limited scale manufacturing of autologous CAR-T, TCR, NK and dendritic cells are obstacles that would need to be overcome before these innovative approaches are widely adopted.

## 5. Other Gene and Cell Therapy Trials

A search for active trials, both recruiting and not recruiting, as well as completed trials, investigating gene and cell therapies specifically for sarcoma were reviewed. The results for active, recruiting clinical trials are shown in Table 1. The results of completed and active not recruiting clinical trials are shown in Table 2.

## 6. Comparisons Between Gene and Cell Therapies and Current Therapies for Cancer

In Table 3, we attempt to show broad comparisons among gene therapy, cell therapy, and currently used chemotherapy, targeted therapy, immunotherapy and radiation therapy. Other cancer therapies including radioablation, cryoablation, hyperthermia, vaccines, repurposed drugs and alternative therapies are not included in this review.

## 7. Conclusions

Gene and cell therapies represent a promising frontier in the treatment of sarcomas, offering targeted therapeutic options going beyond traditional chemotherapy and surgery. In this review, the current landscape of gene and cell therapy clinical trials for sarcomas was assessed, which includes FDA-approved therapies and clinical trials which are actively recruiting, actively not recruiting, and completed studies. The review highlights innovative therapies such as off-the-shelf in vivo tumor-targeted retroviral-based gene therapy (the “Holy Grail” of cancer gene therapy), intratumoral oncolytic virotherapy, CAR-T cell therapy, and natural killer cell therapy, each targeting sarcomas through unique mechanisms.

Gene and cell therapy will soon change the practice of medical oncology. The approval of the first CAR-T cell therapy marks the advent of gene therapy in the field of sarcoma with other CAR-T and TCR cell therapies following closely. Limitations to reaching the tumor and attendant adverse events such as cytokine release syndrome and neurotoxicity can be life-threatening even with the best supportive care. Additional research to optimize CAR-T, TCR and NK cell therapies would include developing personalized CAR-T, TCR and NK cell therapies, and improving genetically modified cell persistence, trafficking and resistance in the TME. Potential solutions would include reducing barriers to entry by reducing extracellular matrix production and consequently enhancing CAR-T, TCR-T and NK cell entry in the TME. The cost of developing these gene-modified T and NK cells is astounding. However, the possibility of curing sarcoma and/or improving the quality of life of patients should necessarily be weighed in.

Expedited development of in vivo targeted injectable vectors such as DNG64 is warranted due to its tumor-agnostic activity in sarcomas and a wide margin of safety. These targeted vectors can easily be combined with metronomic/low-dose chemotherapy, immunotherapy and/or cell therapies to optimize tumor eradication without added toxicity. When given prior to CAR-T, TCR-T or NK cell therapy, DNG64 will kill not only cancer cells and their blood supply, hence reducing tumor burden, but it will also kill stroma-producing fibroblasts, thus facilitating immune cell trafficking in the TME. A brief administration of DNG64 could also reduce the severity of CRS by killing a proportion of immune-activated cells while preserving the efficacy of gene-modified T cells. Economies of scale of vector production in this case would reduce the cost of targeted injectable vectors to approximately that of immune checkpoint inhibitors.

Currently, there are a number of chemotherapies that are known to work in sarcomas, which are usually combined with surgery and radiation therapy. These include doxorubicin, ifosfamide/mesna for all sarcomas, etoposide, vincristine and irinotecan for Ewing sarcoma, gemcitabine, dacarbazine, docetaxel, and trabectedin for soft tissue sarcoma. Targeted therapies and immunotherapy have become increasingly popular for certain subtypes of sarcoma. For sarcomas with specific mutations such as NTRK gene rearrangements, entrectenib and larotrectenib are very effective and are commonly used for sarcomas harboring these mutations and nab-sirolimus is approved for a rare sarcoma known as Perivascular epithelioid cell tumor (PEComa). These drugs are costly as well, but they should also be considered when developing a gene and cell therapy regimen for patients with sarcomas.

Precision medicine has become the dictum of oncology practice. Next-generation sequencing and molecular profiling are diagnostics tools that enable patient-tailored therapies. When designing combinatorial treatment regimens incorporating tumor agnostic versus specific gene and cell therapies, molecularly targeted therapies and chemo/immuno/radiation therapies, clinical oncologists need to consider synergistic activities, associated toxicities, tumor resistance conditions, cost and their effects on the quality of life of patients with sarcoma.

Finally, continued basic, translational and clinical research is essential to optimize gene and cell therapies for sarcomas, and prove their efficacy while reducing risks. The final integration of gene and cell therapies with current chemotherapy, radiation therapy, immunotherapy and targeted therapies will require overcoming the prohibitive cost of manufacturing and improving long-term outcomes and quality of life for patients with this rare and challenging group of cancers.

## Figures and Tables

**Figure 1 cancers-17-01125-f001:**
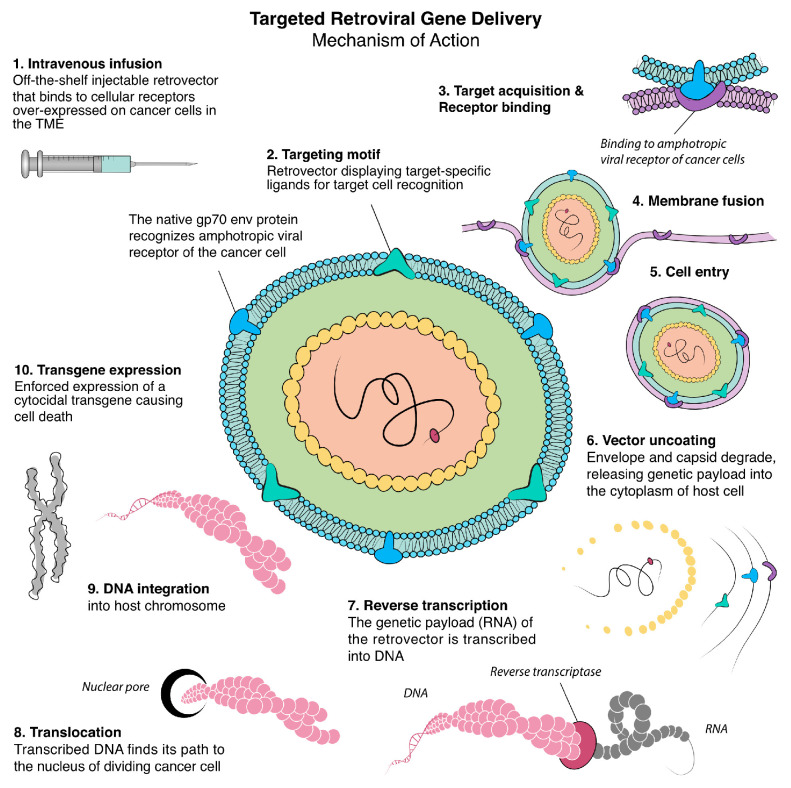
A ten-step illustration of targeted retroviral gene delivery’s mechanism of action. The retroviral nanoparticle displays a target-specific ligand on its glycoprotein 70 (gp70) envelope protein for target cell recognition. This engineered targeted retrovector uses the retroviral vector’s innate property of binding to cellular receptors on mammalian cells, fusing, entering, uncoating, and integrating randomly into the chromosomes of actively dividing cells only. The vector expresses a cytocidal transgene that causes cell death [15].

**Figure 2 cancers-17-01125-f002:**
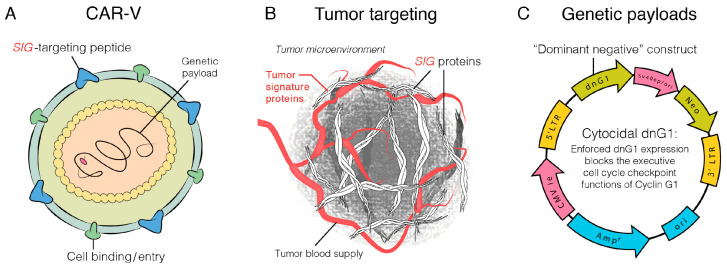
Artist’s illustration of DNG64 vector. The DNG64 Chimeric Amphotropic RNA Vector (CAR-V) displaying a SIG targeting peptide (**A**–**C**), for binding to Signature (Sig) proteins in the tumor microenvironment [TME] (**B**) and encoding a dominant negative human cyclin G1 inhibitor gene (**C**) [21].

**Figure 3 cancers-17-01125-f003:**
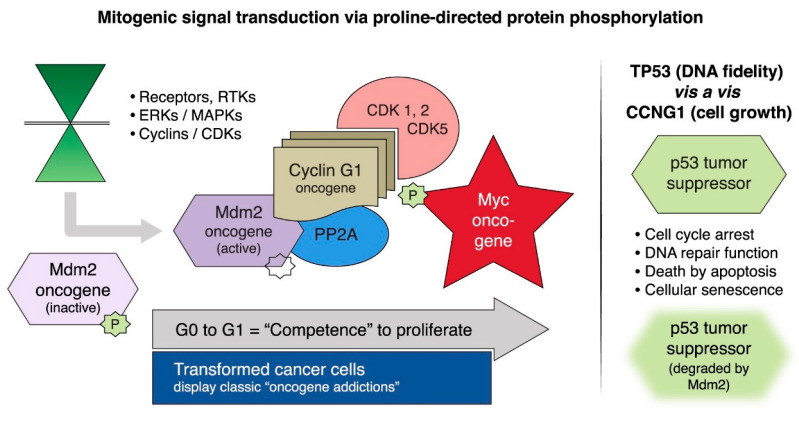
Mitogenic signaling pathways and the human cyclin G1 (*CCNG1*) gene. Left panel: Mitogen-activated (extracellular signaling) protein kinases (MAPKs/ERKs) and cyclin-dependent protein kinase (CDK) complexes control the progressive phases of the cell division cycle. *CCNG1* physically binds to the ser/thr protein phosphatase subunit designated 2A (PP2A) to activate a key regulatory oncoprotein, Mdm2. The Mdm2 oncoprotein forms a physical complex with the p53 tumor suppressor, thus, inactivating its tumor suppressor function, while additionally acting as a specific E3 ubiquitin ligase that is responsible for the ubiquitination and degradation of the p53 tumor suppressor protein. This dephosphorylation event is *CCNG1*-dependent. *CCNG1* also activates Cdk1, CDK5, and CDK2 which can activate the c-Myc onco-protein. Right panel: TP53 tumor suppressor functions are shown as opposed to *CCNG1* growth-promoting function [15].

**Figure 4 cancers-17-01125-f004:**
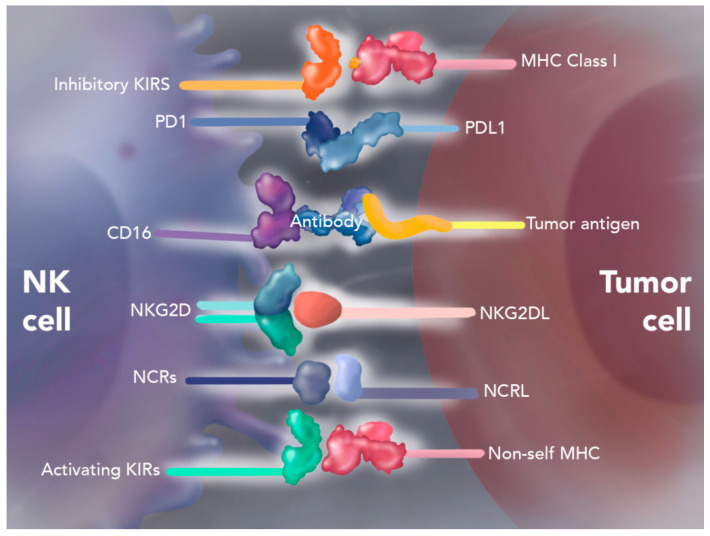
NK cell mechanism of action. NK cells have multiple activating and inhibiting receptors to recognize dangerous cells. The balance between activating and inhibiting signals determines the NK cell’s response. As seen in the graphic, when there is a mismatch between an inhibitory subgroup of KIR receptors on NK cells and major histocompatibility complex (MHC) class I molecules (shown in red) on the surface of target cells, the NK cells can be activated due to lack of inhibitory signals leading to the killing of the target host cell [40].

**Table 1 cancers-17-01125-t001:** Active, recruiting gene and cell therapy trials according to NCT number, status/phase, sarcoma, histologic subtypes, therapy and type of therapy.

NCT Number	Status/Phase	Sarcoma/Histologic Subtype	Therapy	Type of Therapy
NCT06126510	Active, Recruiting Phase 2	Soft Tissue Sarcoma	Recombinant Oncolytic Herpes Simplex Virus Type 1 (HSV-1)	Oncolytic Therapy
NCT06171282	Active, Recruiting Phase 1	Osteosarcoma, Sarcoma, Soft Tissue Sarcoma, Bone Tumor	Recombinant Herpes Simplex Virus I (Oncolytic Virus)	Oncolytic Therapy
NCT05851456	Active, Recruiting Phase 1	Osteosarcoma, Sarcoma, Soft Tissue Sarcoma, Bone Tumor	Recombinant Herpes Simplex Virus I (Oncolytic Virus)	Oncolytic Therapy
NCT05296564	Active, Recruiting Phase ½	Synovial Sarcoma, Soft Tissue Sarcoma, Melanoma Stage IV, Triple Negative Breast Cancer, Metastatic Cancer	Anti-NY-ESO-1 TCR-Gene Engineered Lymphocytes	Gene-Modified Cell Therapy
NCT05621668	Active, Recruiting Phase 1	Soft Tissue Sarcoma, Bone Sarcoma	T-Cell Membrane-Anchored Tumor Targeted IL-12 (Attil-12)	Gene-Modified Cell Therapy
NCT05549921	Active, Recruiting Phase 2	Soft Tissue Sarcoma	TAEST16001 cells (T cells targeting NY-ESO-1)	Gene-Modified Cell Therapy
NCT04897321	Active, Recruiting Phase 1	Ewing Sarcoma, Soft Tissue Sarcoma, Clear Cell Sarcoma, Pediatric Solid Tumor, Osteosarcoma	Autologous T cells genetically engineered to express B7-H3-CARs	Gene-Modified Cell Therapy
NCT04044768	Active, Recruiting Phase 2	Advanced Synovial Sarcoma, Myxoid/Round Cell Liposarcoma	ADP-A2M4 (Afamitresgene autoleucel SPEAR™ T cells)	Gene-Modified Cell Therapy
NCT05642455	Active, Recruiting Phase ½	Synovial Sarcoma, Malignant Peripheral Nerve Sheath Tumor (MPNST), Osteosarcoma	Afamitresgene autoleucel	Gene-Modified Cell Therapy
NCT04995003	Active, Recruiting Phase 1	Sarcoma with HER-2 Overexpression, including Osteosarcoma, Rhabdomyosarcoma, Ewing Sarcoma, Synovial Sarcoma, Soft Tissue Sarcoma, Undifferentiated Sarcoma	HER2-CAR T cells + Immune Checkpoint Inhibitor (Pembrolizumab/Nivolumab)	Gene-Modified Cell Therapy
NCT05607095	Active, Recruiting Phase 1	Undifferentiated Pleomorphic Sarcoma, Dedifferentiated Liposarcoma	Autologous Tumor Infiltrating Lymphocytes (LN-144 or LN-145)	Cellular Therapy
NCT05952310	Active, Recruiting Phase 1/2	Malignant Sarcoma	Allogeneic Haploidentical NK Cells + Chemotherapy/Radiotherapy	Cellular Therapy
NCT04074564	Active, Recruiting Phase 1	Soft Tissue Sarcoma, Osteosarcoma	Multi-antigen Autologous Immune Cell Injection	Cellular Therapy
NCT05634369	Active, Recruiting Phase 1/2	Refractory or Relapsed Pediatric Sarcoma	Infusion of TGFβ Imprinted NK cells	Cellular Therapy

**Table 2 cancers-17-01125-t002:** Completed and active not recruiting clinical trials for sarcoma according to NCT number, status, clinical indication, therapy and type of therapy.

NCT Number	Study Status	Clinical Indication	Therapy	Type of Therapy
NCT02736565	COMPLETED	Metastatic Ewing’s tumor, Ewing Family of Tumors, Recurrent and Advanced Ewing’s Sarcoma, Sarcomas	pbi-shRNA‚ EWS/FLI1 Type 1 Lipoplex	Gene Therapy
NCT05578820	COMPLETED	Sarcoma and other solid tumors	Stimotimagene copolymerplasmid	Gene Therapy
NCT00258687	COMPLETED	Clear cell sarcoma, Alveolar Soft Part Sarcoma, and other solid tumors	GM-CSF gene-transduced tumor vaccine (GVAX)	Gene Therapy
NCT03725605	COMPLETED	Soft Tissue Sarcoma	LTX-315 and Tumor-infiltrating lymphocytes	Oncolytic-peptide Therapy
NCT04318964	ACTIVE_NOT_RECRUITING	Soft Tissue Sarcoma	TCR affinity-enhanced specific T cell therapy (TAEST16001 cells)	Gene-Modified Cell Therapy
NCT02319824	COMPLETED	Sarcoma	Autologous NY-ESO-1-specific CD8-positive T Lymphocytes	Gene-Modified Cell Therapy
NCT03250325	COMPLETED	Synovial Sarcoma	TBI-1301 gene-transduced T cell therapy	Gene-Modified Cell Therapy
NCT01477021	COMPLETED	Liposarcoma, Synovial Sarcoma, Recurrent and Stage III-IV Adult Soft Tissue Sarcoma	NY-ESO-1-specific T cells	Gene-Modified Cell Therapy
NCT03132922	ACTIVE_NOT_RECRUITING	Synovial Sarcoma and Myxoid Round Cell Liposarcoma	Autologous genetically modified MAGE-A4 c1032 T cells	Gene-Modified Cell Therapy
NCT02650986	ACTIVE_NOT_RECRUITING	Advanced Synovial Sarcoma, Unresectable Synovial Sarcoma, and other solid tumors	Autologous NY-ESO-1 TCR/dnTGFbetaRII transgenic T cells	Gene-Modified Cell Therapy
NCT02992743	COMPLETED	Neoplasms	Letetresgene autoleucel (GSK3377794)	Gene-Modified Cell Therapy
NCT05993299	ACTIVE_NOT_RECRUITING	Neoplasms	Letetresgene autoleucel	Gene-Modified Cell Therapy
NCT03462316	ACTIVE_NOT_RECRUITING	Bone Sarcoma, Soft Tissue Sarcoma	NY-ESO-1 TCR Affinity Enhancing Specific T cell Therapy	Gene-Modified Cell Therapy
NCT01241162	COMPLETED	Neuroblastoma, Ewing’s Sarcoma, Osteogenic Sarcoma, Rhabdomyosarcoma, Synovial Sarcoma	Autologous dendritic cell vaccine with adjuvant	Cellular Therapy
NCT00003081	COMPLETED	Sarcoma	Peripheral blood stem cell transplantation	Cellular Therapy
NCT04802070	COMPLETED	Sarcoma	Autologous cytokine-induced killer cells	Cellular Therapy
NCT04052334	COMPLETED	Sarcoma	Tumor-infiltrating lymphocytes	Cellular Therapy
NCT00357396	COMPLETED	Sarcoma	T-Cell Depleted Allogeneic Hematopoietic Stem Cell Transplant	Cellular Therapy
NCT00043979	COMPLETED	Sarcoma	Therapeutic allogeneic lymphocytes, peripheral blood stem cell transplantation	Cellular Therapy
NCT00245011	COMPLETED	Sarcoma	Sm-EDTMP, peripheral blood stem cell transplantation	Cellular Therapy
NCT01347034	COMPLETED	Soft Tissue Sarcoma	Autologous Dendritic Cells	Cellular Therapy
NCT02472392	COMPLETED	Ewing Sarcoma	Allogeneic Stem Cell Transplantation	Cellular Therapy
NCT02100891	COMPLETED	Ewing Sarcoma, Neuroblastoma, Rhabdomyosarcoma, Osteosarcoma, and CNS Tumors	HLA-haploidentical hematopoietic cell transplantation and Donor NK Cell Infusion	Cellular Therapy
NCT02890758	COMPLETED	Soft Tissue Sarcoma, Ewing’s Sarcoma, Rhabdomyosarcoma, and other hematologic and solid tumors	Natural Killer (NK) Cells and ALT803	Cellular Therapy
NCT02849366	COMPLETED	Recurrent Adult Soft Tissue Sarcoma	Natural Killer Immunotherapy	Cellular Therapy
NCT01522820	COMPLETED	Sarcoma and other tumors.	DEC-205/NY-ESO-1 Fusion Protein CDX-1401	Cellular Therapy
NCT00948961	COMPLETED	Advanced Malignancies	CDX 1401 dendritic cell	Cellular Therapy

**Table 3 cancers-17-01125-t003:** Comparisons between gene therapy, cell therapy, chemotherapy, immunotherapy, targeted therapy and radiation therapy.

	Gene Therapy	Cell Therapy	Chemotherapy	Targeted Therapy	Immunotherapy	Radiation Therapy
Mechanism of action	Transfers a cytocidal gene into cancer cells	Transplants healthy cells to enhance immunity, bone marrow recovery	Kills both healthy and normal cells	Targets specific molecules or receptors overexpressed in cancer cells	Stimulates immune system to attack cancer cells	Kills cancer cells using high-energy radiation
Specificity, Yes/No	Yes	Yes	No	Yes	Yes	No
Adverse Reactions/Short-term/Long Term	Short Term: None to minimal (Retroviral-based) to moderate to severe (CRS, neurotoxicity, cytopenias in CAR-T, TCR-T) Long Term: None to Unknown	Short Term: Minimal, infusional reactions to Severe: Cytopenias if conditioning regimens are used Long Term: Second malignancies	Short Term: Moderate to severe bone marrow suppression, organ dysfunction Long Term: Second malignancies, irreversible organ damage	Short Term: Usually mild to moderate, to severe (Interstitial lung disease) Long Term: Unknown	Short Term: Mild to severe, immune-related Long Term: Autoimmune disorders, Irreversible organ	Short Term: Mild to severe organ damage Long Term: Second malignancies Irreversible organ damage
Relative Cost ($)	$$$ to $$$$	$$$	$	$$$	$$$	$$

$$$ to $$$$ (most expensive), $$ (expensive), $ (least expensive).

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
