# Peer review of "Gene and Cell Therapy for Sarcomas: A Review"

_cancers, 2025, doi:10.3390/cancers17071125_

Round 1
Reviewer 1 Report
Comments and Suggestions for Authors
This review by Chawla et. al focuses on the gene and cell therapies for sarcomas that have been approved and therapies still in clinical trials. The reviewers seem to have done a thorough literature review spanning multiple databases including clinicaltrials.gov. Although, the manuscript is timely, well researched and informative, in the current form, the review can benefit from certain elements
1. Section 2 introduction can be edited or shortened and instead a figure can be added. This will help in better understanding of retrovirus-based gene therapy and aid the importance of the section.
2. Sections (2.2 to 2.4) can be subdivided into paragraphs for ease in readability. The sentences are too long. Some of them are 4-5 lines and can be broken as well.
3. It is really encouraging that the authors recognize the increased interest in CAR-T cell-based therapies by providing a detailed section (4.3). However, this section could benefit from a short mechanism of CAR-T therapy along with figure for better understanding of the reader.
4. Overall, the authors should briefly discuss limitations associated with each therapy at the end of each section.
5. The authors have concluded the manuscript well. However, it lacks their perspective of where the field is moving or what should be the areas of research that should potentially be focused on according to them. Although they have touched upon challenges such as limited efficacy in solid tumors, toxicities and high costs; these are important points and should be expanded upon along with cited literature.
Author Response
Reviewer #1
This review by Chawla et. al focuses on the gene and cell therapies for sarcomas that have been approved and therapies still in clinical trials. The reviewers seem to have done a thorough literature review spanning multiple data bases including clinicaltrials.gov. Although, the manuscript is timely, well researched and informative, in the current form, the review can benefit from certain elements:
- Section 2 introduction can be edited or shortened and instead a figure can be added. This will help in better understanding of retrovirus-based gene therapy and aid the importance of the section.
Response: The introduction has been significantly shortened. Figure 1 has been modified to show the 10-step mechanism of action of targeted gene delivery system. See yellow highlighted sections.
- Sections (2.2 to 2.4) can be subdivided into paragraphs for ease in readability. The sentences are too long. Some of them are 4-5 lines and can be broken as well.
Response: Sections 2.2-2.4 have been subdivided into paragraphs and the sentences have been shortened.
- It is really encouraging that the authors recognize the increased interest in CAR-T cell-based therapies by providing a detailed section (4.3). However, this section could benefit from a short mechanism of CAR-T therapy along with figure for better understanding of the reader.
Response: The mechanism of action of CAR-T therapy has been added with a figure (Figure 4). See yellow highlighted section.
- Overall, the authors should briefly discuss limitations associated with each therapy at the end of each section.
Response: An added section on Points to Consider indicating limitations and our envisioned solutions to the limitations. See yellow highlighted sections.
- The authors have concluded the manuscript well. However, it lacks their perspective of where the field is moving or what should be the areas of research that should potentially be focused on according to them. Although they have touched upon challenges such as limited efficacy in solid tumors, toxicities and high costs, these are important points and should be expanded upon along with cited literature.
Response: The discussion section as been re-written indicating our vision and perspectives on where the field is moving toward and areas of research that need to be focused have been added. See yellow highlighted discussion section.
Reviewer 2 Report
Comments and Suggestions for Authors
This is an excellent review by the authors who are internationally well-known experts in sarcoma oncology. The article expertly summarizes the past and current progress of gene and cell therapy for sarcoma, including the most recent presented clinical trial data. This article should be of benefit to many in the field that are not familiar with many of the studies discussed here and help the readers to have better understanding of gene and cell therapy with a historical context.
Author Response
Reviewer #2
This is an excellent review by the authors who are internationally well-known experts in sarcoma oncology. The article expertly summarizes the past and current progress of gene and cell therapy for sarcoma, including the most recent presented clinical trial data. This article should be of benefit to many in the field that are not familiar with many of the studies discussed here and help the readers to have better understanding of gene and cell therapy with a historical context.
Response: We are deeply grateful for this excellent review.
Reviewer 3 Report
Comments and Suggestions for Authors
The review offers a well-rounded and up-to-date look at gene and cell therapies for sarcomas, covering FDA-approved treatments, ongoing clinical trials, and promising new approaches. It does a great job of incorporating numerous clinical studies, providing valuable insights into their progress, effectiveness, and potential impact.
The discussion of molecular mechanisms—particularly for therapies like Rexin-G, DNG64, and Afamitresgene autoleucelis—adds depth and helps readers grasp how these treatments work on a biological level. Additionally, the paper covers a wide range of treatment strategies, including oncolytic virotherapy, CAR-T therapy, NK cell therapy, and dendritic cell vaccines, giving a broad perspective on where the field is heading.
There are a few areas that could be improved-
One major addition would be a direct comparison of gene and cell therapies with traditional treatments like chemotherapy and radiation, which would give readers a clearer picture of how these options stack up. The discussion also tends to focus more on certain therapies, particularly DNG64 and Rexin-G, which might make the review feel a bit one-sided.
While the paper does mention adverse effects, it could go deeper into long-term safety concerns, such as immune-related issues and risks like insertional mutagenesis. Another area for improvement is the inclusion of patient perspectives—covering factors like quality of life and accessibility would make the discussion more well-rounded.
Cost-related challenges are acknowledged, but expanding on regulatory hurdles, reimbursement difficulties, and commercialization barriers would add valuable context.
Lastly, some sections—especially those on clinical trials—repeat similar points, so condensing these would make the paper more concise and easier to follow.
Comments on the Quality of English LanguageWell-structured, clear, and easy to comprehend.
Author Response
Reviewer #3
The review offers a well-rounded and up-to-date look at gene and cell therapies for sarcomas, covering FDA-approved treatments, ongoing clinical trials, and promising new approaches. It does a great job of incorporating numerous clinical studies, providing valuable insights into their progress, effectiveness, and potential impact.
The discussion of molecular mechanisms—particularly for therapies like Rexin-G, DNG64, and Afamitresgene autoleucelis—adds depth and helps readers grasp how these treatments work on a biological level. Additionally, the paper covers a wide range of treatment strategies, including oncolytic virotherapy, CAR-T therapy, NK cell therapy, and dendritic cell vaccines, giving a broad perspective on where the field is heading.
There are a few areas that could be improved-
- One major addition would be a direct comparison of gene and cell therapies with traditional treatments like chemotherapy and radiation, which would give readers a clearer picture of how these options stack up. The discussion also tends to focus more on certain therapies, particularly DNG64 and Rexin-G, which might make the review feel a bit one-sided.
Response: Table 3 shows direct comparisons of gene and cell therapies with treatments currently in use. See yellow highlighted table 3.
While the paper does mention adverse effects, it could go deeper into long-term safety concerns, such as immune-related issues and risks like insertional mutagenesis. Another area for improvement is the inclusion of patient perspectives—covering factors like quality of life and accessibility would make the discussion more well-rounded.
Response: The sections on Points to Consider have been added at the end of each section. Long term safety concerns about gene and cell therapies for sarcomas have been added and patient perspectives regarding quality of life and accessibility have been added in the Discussion section. See yellow highlighted sections.
Cost-related challenges are acknowledged, but expanding on regulatory hurdles, reimbursement difficulties, and commercialization barriers would add valuable context.
Response: Challenges of gene and cell therapies, regulatory hurdles, reimbursement issues and commercialization barriers have been added in the Discussion section. See yellow highlighted Discussion section.
Lastly, some sections—especially those on clinical trials—repeat similar points, so condensing these would make the paper more concise and easier to follow.
Response: Sections on clinical trials that repeat similar points have been deleted. Additionally, the clinical trials have been organized to reflect retroviral based gene trials, oncolytic trials and cell therapy trials. See modified Tables 1 and 2 which are yellow highlighted.
Reviewer 4 Report
Comments and Suggestions for Authors
This is a well-researched review with strong potential, but it would benefit from improved organization, a more detailed discussion of clinical trials, and a critical evaluation of implementation challenges.
- While the review mentions clinical trials, it does not provide a detailed analysis of their outcomes, limitations, or the challenges of moving from early-stage to late-stage trials.
- Although the review is optimistic, it would benefit from discussing barriers to implementation, such as immune-related toxicities of CAR-T therapies, logistical challenges in virotherapy, and high costs of gene therapy.
-Addressing resistance mechanisms in sarcomas to these therapies would add depth.
- While the review discusses therapy approvals, it lacks an in-depth discussion of the molecular targets and pathways involved in gene and cell therapies for sarcomas.
Author Response
Reviewer #4
- This is a well-researched review with strong potential, but it would benefit from improved organization, a more detailed discussion of clinical trials, and a critical evaluation of implementation challenges.
Response: The discussion section has been re-written and the Tables on Clinical Trials have been organized to reflect retroviral based gene trials, oncolytic trials and cell therapy trials. See modified Tables 1 and 2 which are yellow highlighted.
- While the review mentions clinical trials, it does not provide a detailed analysis of their outcomes, limitations, or the challenges of moving from early-stage to late-stage trials.
Response: There are only a 3 gene therapy products for sarcoma that have been approved by a regulatory agency, and we have discussed/analyzed in detail those clinical trials that have promising results. We have also discussed the limitations and challenges of moving a gene therapy product from early stage to late stage trials. If we have missed any promising clinical trial data, please let us know and we will quickly add them to the review article. See yellow highlighted sections.
- Although the review is optimistic, it would benefit from discussing barriers to implementation, such as immune-related toxicities of CAR-T therapies, logistical challenges in virotherapy, and high costs of gene therapy.
Response: Points to consider sections have been added to address these issues and concerns. Also Table 3 has been added to compare gene and cell therapies with current therapies in use for sarcoma. The discussion section has been re-written to summarize these issues as well. Please refer to yellow highlighted sections.
- Addressing resistance mechanisms in sarcomas to these therapies would add depth.
Response: Resistance mechanisms in sarcomas to gene and cell therapies have been added. See yellow highlighted sections.
- While the review discusses therapy approvals, it lacks an in-depth discussion of the molecular targets and pathways involved in gene and cell therapies for sarcomas.
Response: Molecular targets and pathways involved in gene and cell therapies for sarcomas are now provided. See yellow highlighted sections.